# Towards the Resolution of a Quantized Chaotic Phase-Space: The Interplay of Dynamics with Noise

**DOI:** 10.3390/e25030411

**Published:** 2023-02-24

**Authors:** Domenico Lippolis, Akira Shudo

**Affiliations:** 1Institute for Applied Systems Analysis, Jiangsu University, Zhenjiang 212013, China; 2Department of Physics, Tokyo Metropolitan University, Minami-Osawa, Hachioji 192-0397, Tokyo, Japan

**Keywords:** quantum dissipation, stochasticity, nonlinearity, chaos, Wigner equation, Fokker–Planck equation

## Abstract

We outline formal and physical similarities between the quantum dynamics of open systems and the mesoscopic description of classical systems affected by weak noise. The main tool of our interest is the dissipative Wigner equation, which, for suitable timescales, becomes analogous to the Fokker–Planck equation describing classical advection and diffusion. This correspondence allows, in principle, to surmise a finite resolution, other than the Planck scale, for the quantized state space of the open system, particularly meaningful when the latter underlies chaotic classical dynamics. We provide representative examples of the quantum-stochastic parallel with noisy Hopf cycles and Van der Pol-type oscillators.

## 1. Introduction

Efforts to reconcile classical and quantum mechanics are just about as old as quantum mechanics itself. While the formulation in Hilbert space makes it difficult to establish a direct correspondence between the two, a projection of the wave function to phase space may reveal some formal affinities between the quantum evolution of probability density and the traditional Liouville formalism of classical mechanics. The closest one can get to relate the two is by projecting the Liouville–von Neumann equation onto a suitable state space. For example, choosing the traditional phase space, we may obtain the so-called Wigner representation, which shares similarities with the aforementioned classical density evolution.

Yet, there are also notable differences, as it stands to reason. The Wigner function, which is the projection of the density operator onto the phase space, may also take on negative values; its evolution is governed by an equation plagued with an infinity of derivatives, and, as an indirect consequence of that, it may attain scales smaller than Planck’s constant [1]. This is especially true in systems whose underlying classical dynamics exhibit chaotic behavior.

In reality, however, no system is perfectly and eternally isolated, and exchanges of matter or energy with the surrounding environment are inevitable, whether due to measurements, thermal interactions, or shot noise [2,3,4]. That brings dissipation into the picture and, with that, decoherence.

The effect of the environment on the evolution of a density matrix in a phase-space representation was first studied by Feynman and Vernon [5], who extended the path-integral formalism to dissipative quantum dynamics. Later, Caldeira and Leggett [6] derived an equivalent partial differential equation for the density matrix, which bears diffusive and dissipative terms similar to the Fokker–Planck equation. The latter analogy was then thought to hold in the semiclassical limit until a new wave of contributions [7,8,9,10] reexamined the problem in a quantum chaotic setting. A most remarkable outcome of those works is the identification of a decoherence time, beyond which the Wigner equation is, in all, a Fokker–Planck equation since the higher-order derivatives may be safely neglected, and the quantized phase space may be resolved only up to a finite scale. Such resolution does not depend on the Planck constant, but rather emerges from the balance of the phase-space contraction rate (Lyapunov exponent) with the coupling of the system with an Ohmic environment.

More recent contributions have focussed on the efficiency of Wigner evolution for general types of dissipation [11], and on obtaining a Lindblad-based dissipative Wigner equation to tackle quantum friction [12,13].

Once it is established that, under suitable conditions and after a sufficiently long time of evolution, the Wigner equation has the form of a Fokker–Planck equation, the quantum dissipative problem is cast into a classical stochastic process. Moreover, if the underlying classical dynamics of the quantum system in exam are chaotic, the limiting resolution of the phase space postulated in refs. [8,10] is not expected to be uniform, but will depend on the local interplay of the stretching/contraction with the dissipation. In the equivalent classical noisy problem, it is the ‘Brownian’ diffusion that plays the role of dissipation.

In the past decade, significant steps [14,15,16,17,18] were taken to determine the resolution of a chaotic state space in the presence of weak noise and reduce the dynamics to a Markov process of finite degrees of freedom in the form of a connectivity matrix. Low-dimensional discrete-time dynamical systems such as logistic- or Hénon-type maps have been treated in a non-Hamiltonian setting, whose quantum analogs are, in principle, difficult to identify. The optimal resolution hypothesis should be extended to continuous time flows as well, and the starting point of that roadmap is a thorough comprehension of the steady-state solutions of the Fokker–Planck equation around the building blocks of chaos: periodic orbits.

Here, we intend to lay the foundation of that understanding, by solving the Fokker–Planck equation of nonlinear paradigmatic dynamical systems, classical and with weak noise. We examine two-dimensional flows featuring nonlinearities but not yet chaos, where the competition between contraction and noise around a limit cycle results in a stationary density, which characterizes the steady state, and, as shown at the very end of the manuscript, shares common traits with the steady-state Wigner function of a case study in quantum dissipative dynamics.

The article is structured as follows: in Section 2 we review the basic tools of the phase-space representation of quantum dynamics, both in closed and open systems. We follow up in Section 3 by discussing the main issues related to the evolution of the Wigner function in a quantum chaotic setting, the effects of dissipation, and the correspondence of the Wigner with the Fokker–Planck equation. In Section 4, a novel methodology is introduced to evaluate the steady-state solution of the Fokker–Planck equation around a periodic orbit, which casts the partial differential equation into an ordinary differential equation for the covariance matrix, known as the Lyapunov equation. We first present a proof of concept on the simplest limit cycle, of circular shape as from a Hopf bifurcation, to be followed in Section 5 by the treatment of the nonlinear oscillators, which are the main object of the current study. At the end of the section, the results on the Fokker–Planck steady-state densities are paralleled to those obtained for the Wigner function in a recent study of a quantum-dissipative model of the same oscillators. Summary and discussion close the paper.

## 2. Density Matrix, Wigner Function, and Dissipation

Given a collection of physical systems, the ensemble average of an observable *A* is given by
(1)$$\langle A\rangle =\sum_{i}\rho_{i}\langle \psi_{i},\left|A\right|,\psi_{i}\rangle ,$$
or, using
(2)$$\rho =\sum_{i}\rho_{i}|\psi_{i}\rangle \langle \psi_{i}|\;,$$
one can simply write
(3)$$\langle A\rangle =\mathrm{Tr}\left[\rho \;A\right]\;,$$
so that, if the observable *A* is time-independent, knowing ρ at all times means solving the problem of dynamics. That is the motivation for studying the density matrix ρ in the first place.

### 2.1. Quantum Dynamics in the Phase Space

Now, the density ρ evolves according to the Liouville–von Neumann equation
(4)$$i\hbar \rho_{t}=\left[\rho ,H\right]\;,$$
the quantum analog of the well-known Liouville equation
(5)$$\rho_{t}=\left\{\rho ,H\right\}$$
of classical dynamics, which we spell out in phase space: (6)$$\partial_{t}\rho =-\frac{p}{m}\partial_{x}\rho +\left(\partial_{x}V\left(x\right)\partial_{p}\right)\rho \;,$$
assuming a Hamiltonian of the form $H=\frac{p^{2}}{2m}+V\left(x\right)$.

In order to integrate the Liouville–von Neumann Equation (Equation 4), we need to project it onto some basis, and several representations are already available to us, for instance, the P- or the *Q*-representation (a.k.a. Husimi’s)
(7)$$Q\left(\alpha ,\alpha^{*}\right)=\frac{1}{\pi }\langle \alpha |\rho |\alpha \rangle \;,$$
with |α〉 a coherent state. Studying quantum-to-classical correspondence, especially of a system that exhibits chaotic behavior, is generally best achieved by using the Wigner representation [19]
(8)$$W\left(x,p\right)=\frac{1}{2\pi \hbar }\int e^{-ipy/\hbar }\psi \left(x+\frac{y}{2}\right)\psi^{*}\left(x-\frac{y}{2}\right)dy\;,$$
which can also be expressed in terms of the density matrix, as
(9)$$W\left(x,p\right)=\frac{1}{2\pi \hbar }\int e^{-ipy/\hbar }\langle x+y/2|\rho |x-y/2\rangle dy\;,$$
the operation being called Weyl transform. An operator *A* may also be projected onto phase space, by applying a Weyl transform: (10)$$\tilde{A}\left(x,p\right)=\frac{1}{2\pi \hbar }\int e^{-ipy/\hbar }\langle x+y/2|A|x-y/2\rangle dy\;,$$
which can prove handy in the evaluation of expectation values, that is
(11)$$\langle A\rangle =\mathrm{Tr}\left[\rho \;A\right]=\int W\left(x,p\right)\tilde{A}\left(x,p\right)dxdp\;,$$
since, in general,
(12)$$\mathrm{Tr}\left[AB\right]=\int \tilde{A}\left(x,p\right)\tilde{B}\left(x,p\right)dxdp\;.$$ Thus, expectation values of observables are determined by means of phase-space averages, and the problem of quantum mechanics boils down to that of the time evolution of the Wigner function. It has been shown [19] that W(x,p) obeys the Wigner equation
(13)$$\partial_{t}W\left(x,p\right)=-\frac{p}{m}\partial_{x}W\left(x,p\right)+\sum_{s=0}^{\infty }c_{s}{\left(-\hbar^{2}\right)}^{s}\partial_{x}^{2s+1}V\left(x\right)\partial_{p}^{2s+1}W\left(x,p\right)\;,$$ (with $c_{s}=\frac{2^{-2s}}{(2s+1)!}$) that, in general, bears an infinite number of terms. In reality, integrating Equation (Equation 13) can already be impractical if there are just a few nontrivial terms in the summation [20]. If the potential V(x) is at most quadratic, the Wigner equation reduces to Liouville’s, as in (Equation 6). Otherwise, Equation (Equation 13) is still not easy to deal with, and, importantly, it may not be truncated in the semiclassical limit since the terms $\partial_{p}^{2s+1}W\left(x,p\right)$ bring down powers of $\hbar^{-1-2s}$, so that
$$\hbar^{2s}\cdot \frac{1}{\hbar^{2s+1}}∼\hbar^{-1}\;,$$
and $O\left(\hbar^{-1}\right)$ does grow in the limit ℏ→0, making no terms in the Wigner equation negligible, in principle.

### 2.2. Open Systems

On the other hand, let us suppose the system is connected to an environment, whose interaction produces two additional terms on the right-hand side of the Wigner Equation (Equation 13), that is [8]
$$2\gamma \partial_{p}\left[pW(x,p)\right]+D\partial_{pp}^{2}W\left(x,p\right)\;.$$ The first term produces relaxation, due to the exchange of energy with the environment, and γ is the relaxation rate. The second term means diffusion, responsible for the so-called decoherence process, where one sets $D=2\gamma Mk_{B}T$, with *M* mass of the system, and *T* temperature of the environment. The dissipation and diffusion terms are obtained from a path-integral formulation of the system-environment interaction, which traces back to the works of Feynman and Vernon [5], and, later, of Caldeira and Leggett [6].

If the potential V(x) is, at most, quadratic, one recovers the Fokker–Planck equation, which describes the classical evolution of the density of trajectories produced by a particle subject to Brownian motion: (14)$$\partial_{t}W\left(x,p\right)=-\frac{p}{m}\partial_{x}W\left(x,p\right)+\partial_{x}V\left(x\right)\partial_{p}W\left(x,p\right)+2\gamma \partial_{p}\left[pW(x,p)\right]+D\partial_{pp}^{2}W\left(x,p\right)\;.$$ This equation is fully quantum mechanical, and W(x,p) may take on negative values, unlike the classical phase-space density of a Brownian particle.

Yet, for a general potential V(x), the evolution of the dissipative system is ruled by the full-fledged Wigner Equation (Equation 13) plus the terms (Equation 14) due to the environment: (15)$$\begin{array}{c} \partial_{t}W\left(x,p\right)=-\frac{p}{m}\partial_{x}W\left(x,p\right)+\sum_{s=0}^{\infty }c_{s}{\left(-\hbar^{2}\right)}^{s}\partial_{x}^{2s+1}V\left(x\right)\partial_{p}^{2s+1}W\left(x,p\right)+\\ +\;2\gamma \partial_{p}\left[pW(x,p)\right]+D\partial_{pp}^{2}W\left(x,p\right)\;. \end{array}$$ The resulting equation is still plagued with an infinite number of derivatives, and is thus of impractical integration. In the next section, we discuss whether and how it is safe to neglect the higher-order terms in Equation (Equation 15), in the context of quantum chaos.

## 3. Stretching, Contracting, and Zaslavsky’s Time

Let us examine some aspects of the evolution of the Wigner function, when the underlying classical dynamics of the system is chaotic. By established knowledge [21], the two main features of chaos are:Nearby trajectories diverge exponentially fast, meaning that, letting x=(x,p),
(16)$$\lambda =\lim_{t\to \infty }\ln\left|\frac{\delta x(t)}{\delta x(0)}\right|>0\;,$$
in other words, the difference δx(t) between any two nearby trajectories grows exponentially fast for any initial condition. This feature is also described as extreme sensitivity of the system to initial conditions.The number *M* of qualitatively distinct orbits (‘configurations’, tagged by symbolic sequences) scales exponentially with their length, so that the topological entropy is positive:
(17)$$S=\lim_{t\to \infty }\frac{1}{t}\ln M\left(t\right)>0\;.$$

### 3.1. Chaos and the Wigner Function

Chaos is the result of a stretching and folding process mainly due to nonlinearities. For a Hamiltonian system, volumes in the phase space are conserved (by Liouville’s theorem), so that the amount of stretching (diverging trajectories) in some directions must be compensated by an equal amount of contraction in others.

As a result, the inconvenient higher-order terms in the Wigner Equation (Equation 13) can be estimated to evolve as
(18)$$\partial_{p}^{2s+1}W\left(x,p\right)\propto \frac{W(x,p)}{\delta p^{2s+1}\left(t\right)}∼\frac{W(x,p)}{\delta p\left(0\right)\;e^{-(2s+1)\lambda t}}\;,$$
for smooth enough W(x,p). Thus, the inherent problem is, in principle, not with the smoothness of the density, but rather with the fact that the phase space contracts at an exponential rate, and therefore the contribution of higher-order derivatives in Equation (Equation 13) is more and more important, as time proceeds. To better illustrate that, let us compare the terms in the Poisson brackets of the Liouville Equation (Equation 6) (also present in the full-fledged Wigner equation), with the higher-order terms in Equation (Equation 13): (19)$$\frac{\partial_{x}V\left(x\right)\partial_{p}W\left(x,p\right)}{c_{s}\partial_{x}^{2s+1}V\left(x\right)\partial_{p}^{2s+1}W\left(x,p\right)}∼\frac{1}{c_{s}}\frac{\partial_{x}V\left(x\right)}{\partial_{x}^{2s+1}V\left(x\right)}\delta p^{2s}∼\frac{1}{c_{s}}\frac{\partial_{x}V\left(x\right)}{\partial_{x}^{2s+1}V\left(x\right)}\delta p\left(0\right)\;e^{-2s\lambda t}\gg 1\;$$
is a condition for the higher-order terms to be negligible with respect to the lower-order ‘Liouville’ terms. The above inequality can be inverted, and turned into a condition for the time *t*: (20)$$t\ll \frac{1}{\lambda }\ln{\left[\frac{\partial_{x}V\left(x\right)}{\partial_{x}^{2s+1}V\left(x\right)}\frac{\delta p(0)}{c_{s}}\right]}^{-1/2s}\;.$$ Identifying the quantity $X_{V}\;\delta p\left(0\right)=\frac{\partial_{x}V\left(x\right)\delta p\left(0\right)}{\partial_{x}^{2s+1}V\left(x\right)}$ with the typical action of the system, we can now understand
(21)$$t^{*}∼\frac{1}{\lambda }\ln\frac{X_{V}\;\delta p\left(0\right)}{\hbar }$$
as the time scale within which the inconvenient higher-order terms (s≥1) of the Wigner Equation (Equation 13) may be neglected. Some literature refers to $t^{*}$ as Zaslavsky’s time [22]. Its meaning is somehow related to the more commonly mentioned Ehrenfest time, as in fact, $t^{*}$ is longer the larger the ratio of the typical action to *ℏ*, and longest in the semiclassical limit. Still, the basic idea of this correspondence time does not relate directly with interference or need ‘semiclassical’ dynamics, but rather implies a finite resolution for the quantized phase space within a certain time scale, irrespective of the scale of the action. In general, the smoother the potential V(x), the longer $t^{*}$, whereas the larger the Lyapunov exponent λ, the shorter $t^{*}$.

### 3.2. A Resolution for the Quantized Phase Space

In a simplified but physically meaningful description, that will then prove more accurate as a local model, we may recognize and estimate the competing effects of dynamical contraction on the one hand, and of dissipation-induced diffusion on the other hand, by quantizing the Hamiltonian H=λxp. A wave packet of the form
(22)$$W\left(x,p\right)∼e^{-x^{2}/\sigma^{2}-\sigma^{2}p^{2}}$$
evolves separately along the stretching *x*-direction, and the contracting *p*-direction. In momentum space, we have the Schrödinger equation
(23)$$\partial_{t}u\left(p,t\right)=\lambda p\partial_{p}u\left(p,t\right)\Rightarrow u\left(p,t\right)=u_{0}\left(pe^{\lambda t}\right)\;,$$
that maps the wave packet in the *p*-direction as
(24)$$e^{-\sigma^{2}p^{2}/2}\to e^{e^{2\lambda t}\sigma^{2}p^{2}/2}\;,$$
and thus the width $\sigma^{-2}$ shrinks by a factor of $e^{-2\lambda t}$ after time *t*. Identifying $\sigma^{-1}$ with the uncertainty δp(t) of the momentum, we may say that
(25)$$\delta p\left(t\right)∼\delta p\left(0\right)e^{-\lambda t}$$
along the contracting direction. On the other hand, connecting the system to an environment brings about diffusion, and
(26)$$\partial_{t}u\left(p,t\right)=D\partial_{pp}u\left(p,t\right)\Rightarrow u\left(p,t\right)∼e^{-p^{2}/2\left(\delta p\left(0\right)+Dt\right)}\;,$$
whose variance evolves as $\sqrt{Dt}$: (27)$$\delta p\left(t\right)∼{\left[\delta p(0)+Dt\right]}^{1/2}\;.$$ Then, intuitively, there must be some minimal scale in the contracting direction, set by
(28)$$\delta p_{\min}∼{\left[\frac{D}{2\lambda }\right]}^{1/2}\;.$$ The full picture is called the Ornstein–Uhlenbeck problem [23]
(29)$$\partial_{t}u\left(p,t\right)=D\partial_{pp}u\left(p,t\right)-\lambda \partial_{p}u\left(p,t\right)\;.$$ In particular, the larger $\delta p_{\min}$, the closer the evolution of the Wigner function to a stochastic process. More precisely, the regime where we may neglect the higher-order derivatives in the Wigner equation is deduced from Equation (Equation 19) as
(30)$$\frac{X_{V}\;\delta p_{\min}}{\hbar }\gg 1\;,$$
and that requires a relatively smooth potential, and the coefficient of the decoherence term in Equation (Equation 14), *D*, to be comparable to the Lyapunov exponent λ. Now, if $\delta p_{\min}$ is an ‘equilibrium’ value as argued above, the chaotic contraction is no longer shrinking the scale of phase-space probability exponentially and indefinitely as in the non-dissipative setting (recall $\delta p\left(t\right)∼\delta p\left(0\right)e^{-\lambda t}$). Hence, in principle, there would be no Zaslavsky’s time $t^{*}$, but rather, the quantum dissipative evolution may be well described by the Fokker–Planck type of Equation (Equation 14) at all times, provided that the initial condition is smooth enough. Importantly, the semiclassical limit is not required for this approximation to work.

## 4. Contraction vs. Diffusion in Stochastic Dynamics

Equation (Equation 14) and the discussion from the previous section suggest that, under suitable conditions, the problem of the dynamics of a quantum system connected to an environment may be cast into the classical evolution of a density according to a Fokker–Planck equation. As a consequence, studying the interplay of stretching/contracting dynamics with weak noise may also help shed light on quantum dissipation. Particularly interesting scenarios arise when the deterministic dynamics exhibits chaotic behavior. It is, in fact, well known that the phase space of a chaotic system has a self-similar (fractal) structure of infinite resolution. However, in reality, every system experiences noise, coming from experimental uncertainties, neglected degrees of freedom, or roundoff errors, for example. No matter how weak, noise smooths out fractals, giving the system a finite resolution. The consequences are dramatic for the computation of long-time dynamical averages, such as diffusion coefficients or escape rates, since infinite-dimensional operators describing the evolution of the system (such as Fokker–Planck) effectively become finite matrices. With the aim of efficiently estimating long-time averages of observables for a chaotic dynamical system affected by background noise, a recent endeavor carried on over the past decade has achieved a technique to partition the chaotic phase space up to its optimal resolution, using unstable periodic orbits. The benchmark models already treated range from one-dimensional discrete-time repellers [14], and general unimodal maps [15], to two-dimensional chaotic attractors [17,18]. Most importantly, a finite resolution for the state space of these models has effectively changed the dimensionality of the Fokker–Planck operator from infinite to inherently finite. Consequently, computations of the desired long-time averages become simpler and more efficient. On a more intuitive note, the present results also bear physical significance because, even when the external noise is uncorrelated, additive, isotropic, and homogenous, the interplay of noise and nonlinear dynamics always results in a local stochastic neighborhood, whose covariance depends on both the past and the future noise integrated and nonlinearly convolved with deterministic evolution along the trajectory. In that sense, noise is effectively never ‘white’ in nonlinearity, and thus, the optimal resolution varies from neighborhood to neighborhood and has to be computed locally.

As stated in the introduction, here we attack continuous-time dynamical systems, and begin by studying the evolution of noisy neighborhoods of periodic orbits. The simplest yet meaningful models are two-dimensional limit cycles, that can serve as a testbed for parsing the interaction of deterministic dynamics with noise.

### 4.1. The Lyapunov Equation around a Cycle

Consider the Fokker–Planck equation
(31)$$\partial_{t}\rho \left(x,t\right)=-\partial_{x}\left(v(x)\rho (x,t)\right)+\Delta \partial_{xx}\rho \left(x,t\right)\;,$$
where Δ is the diffusion tensor, whose entries are the noise amplitudes along each direction (Δ is diagonal with identical entries for isotropic noise). If we look at the dynamics in the neighborhood of a particular deterministic trajectory, we may linearize the velocity field v(x) locally, and replace it with $A_{a}\left(x-x_{a}\right)$, where $A_{a}=\frac{\partial v(x)}{\partial x}{|}_{x=x_{a}}$ is the so-called matrix of variations. Moreover, we may switch to a co-moving reference frame in the desired neighborhood, say $z_{a}=x-x_{a}$.

Suppose we start off with an initial density of trajectories of Gaussian shape, that is $\rho_{a}\left(z_{a}\right)=\frac{1}{C_{a}}\exp\left(-z{}_{a}^{⊤}\frac{1}{Q_{a}}\;z_{a}\right)$. The short-time solution to (Equation 31) can then be written in the path-integral form
(32)$$\begin{array}{ccc} \rho_{a+1}\left(z_{a+1}\right) & = & \frac{1}{C_{a}}\int \left[dz_{a}\right]\;e^{-\frac{1}{2}\left(z_{a+1}-\left(1+A_{a}\delta t\right)z_{a}\right){}^{⊤}\frac{1}{\Delta \delta t}\;\left(z_{a+1}-\left(1+A_{a}\delta t\right)z_{a}\right)-\frac{1}{2}z{}_{a}^{⊤}\frac{1}{Q_{a}}\;z_{a}}\\ & = & \frac{1}{C_{a+1}}\;e^{-\frac{1}{2}z{}_{a+1}^{⊤}\frac{1}{\left(1+A_{a}\delta t\right)Q_{a}{\left(1+A_{a}\delta t\right)}^{T}+\Delta \delta t}\;z_{a+1}}\;. \end{array}$$ One can then infer the relation between input and output quadratic forms in the exponential
(33)$$Q_{a+1}=\Delta \delta t+\left(1+A_{a}\delta t\right)Q_{a}\left(1+A_{a}\delta t\right){}^{⊤}\;,$$
and, neglecting terms of order $\delta t^{2}$, recover the time-dependent Lyapunov equation
(34)$$\dot{Q}=A\left(t\right)Q+QA{}^{⊤}\left(t\right)+\Delta \;,\;Q\left(t_{0}\right)=Q_{0}\;.$$

Following the theory of time-dependent ordinary differential equations [24], we may write the solution of (Equation 34) as
(35)$$Q\left(t\right)=J\left(t,t_{0}\right)\left[Q\left(t_{0}\right)+\int_{t_{0}}^{t}J^{-1}\left(s,t_{0}\right)\Delta \left(J^{-1}\left(s,t_{0}\right)\right){}^{⊤}ds\right]J{}^{⊤}\left(t,t_{0}\right)\;.$$ Here, $J(t,t_{0})$ is the Jacobian along a flow x=x(t):(36)$$\frac{d}{dt}J\left(t,t_{0}\right)=A\left(x\right)J\left(t,t_{0}\right),\;J\left(t_{0},t_{0}\right)=1\;.$$ One can verify this by just plugging the solution above into the equation. Alternatively, one can write Equation (Equation 35) in the simpler form
(37)$$Q\left(t\right)=J\left(t,t_{0}\right)Q\left(t_{0}\right)J{}^{⊤}\left(t,t_{0}\right)+\int_{t_{0}}^{t}J\left(t,s\right)\Delta J{}^{⊤}\left(t,s\right)ds\;,$$
where the notation J(t,s) means that the Jacobian is computed following a trajectory that starts at time *s* and ends at time *t*, consistently with Equation (Equation 36).

### 4.2. Noisy Circle

Next, consider one of the simplest two-dimensional dynamical systems, a pair of ODEs with a circular limit cycle of radius $r_{c}$, together with additive isotropic white noise of strength 2D:(38)$$\begin{array}{ccc} \dot{x} & = & \lambda \left(r_{c}-\sqrt{x^{2}+y^{2}}\right)x-\omega y+\sqrt{2D}\xi_{x}\\ \dot{y} & = & \lambda \left(r_{c}-\sqrt{x^{2}+y^{2}}\right)y+\omega x+\sqrt{2D}\xi_{y} \end{array}$$
where
(39)$$<\xi_{x}\left(t\right)\xi_{x}\left(\tau \right)>=\delta \left(t-\tau \right),\;<\xi_{x}\left(t\right)\xi_{y}\left(\tau \right)>=0.$$ In polar coordinates, this is written
(40)$$\begin{array}{ccc} \dot{r} & = & \lambda \left(r_{c}-r\right)r+\sqrt{2D}\xi_{x}\cos\theta +\sqrt{2D}\xi_{y}\sin\theta \\ \dot{\theta } & = & \omega -\sqrt{2D}\xi_{x}\frac{\sin\theta }{r}+\sqrt{2D}\xi_{y}\frac{\cos\theta }{r} \end{array}$$ This Langevin-type equation produces the drift and diffusion coefficients [23]
(41)$$\begin{array}{ccc} D_{r} & = & \lambda \left(r_{c}-r\right)r+\frac{2D}{r}\\ D_{\theta } & = & \omega \\ D_{rr} & = & 2D\\ D_{\theta \theta } & = & \frac{2D}{r^{2}} \end{array}$$
which then determine the Fokker–Planck equation for the system:(42)$$\partial_{t}P+\frac{1}{r}\partial_{r}\left[\lambda \left(r_{c}-r\right)rP\right]+\partial_{\theta }\omega P-\frac{D}{r}\partial_{r}\left(r\partial_{r}P\right)-\frac{D}{r^{2}}\partial_{\theta \theta }P=0$$ The limit cycle $r=r_{c}$ can be either stable (Figure 1) or unstable depending on the sign of λ. Let us consider the stable case.

The first thing to look for is a stationary solution to the asymptotic form of (Equation 42):(43)$$\partial_{r}\left[\lambda \left(r_{c}-r\right)rP_{\infty }\right]-D\partial_{r}\left(r\partial_{r}P_{\infty }\right)=0$$ A solution is
(44)$$P_{\infty }\left(r\right)=Ce^{-\frac{\lambda }{2D}{\left(r-r_{c}\right)}^{2}}\;,$$
which implies that $P_{\infty }$ is a Gaussian of width $2\sqrt{D/\lambda }$ in the neighborhood of the limit cycle. The general solution to (Equation 42) is [23]
(45)$$P\left(r,\theta ,t\right)=e^{-\frac{\lambda }{2D}{\left(r-r_{c}\right)}^{2}}\sum_{n=0}^{\infty }\sum_{\nu =-\infty }^{\infty }A_{n}^{\nu }e^{-s_{n}^{\nu }t}{\left(r-r_{c}\right)}^{\left|\nu \right|}L_{n}^{\left|\nu \right|}\left(r-r_{c}\right)e^{i\nu \theta }$$
where $L_{n}^{\left|\nu \right|}\left(r-r_{c}\right)$ are generalized Laguerre polynomials and both the eigenvalues $s_{n}^{\nu }$ and the coefficients $A_{n}^{\nu }$ can be found numerically.

#### Neighborhood and Coordinates

This problem has an obvious symmetry, which allows us to guess the right (nonlinear!) change of coordinates, as well as the stationary solution, independent of the angular coordinate. The result is that the noisy neighborhood of the limit cycle is determined by the variance of the stationary solution (Equation 44). In general, however, we might not be so lucky, and guessing a suitable, possibly nonlinear, change of coordinates is probably beyond our reach. One way to identify a neighborhood for a periodic or any other orbit is to integrate the time-dependent Lyapunov Equation (Equation 34) in the original (Cartesian) coordinates, but in a co-moving frame defined by the local coordinates $z_{a}=x-x_{a}$ introduced in Section 4.1.

Figure 2 illustrates this second approach: the forward Lyapunov Equation (Equation 34) is numerically solved along an orbit that converges to the circular limit cycle (here we take the diffusion tensor $\Delta =\left(\begin{array}{cc} 2D & 0\\ 0 & 2D \end{array}\right)$), and its solution (Equation 37) is sampled along the trajectory and *inverted* to obtain $Q^{-1}\left(t\right)$, the covariance matrix of the Gaussian density, that produces a tube (in the figure in light yellow) along the orbit. The eigenvalues of $Q^{-1}\left(t\right)$ are found to converge to $\Lambda_{1}=\frac{\lambda }{2D}$, consistently with the result for the width of the stationary-state solution (Equation 44) of the full-blown radial Fokker–Planck Equation (Equation 43): that determines the width of the tube, $\sigma =\sqrt{1/2\Lambda_{1}}$. The second eigenvalue of $Q^{-1}$ is $\Lambda_{2}=0$, as it appears from Figure 2b. The latter eigenvalue is to be read as follows: while the forward Lyapunov equation converges to a finite limit in the stable (radial) direction, where noise balances contracting dynamics, it diverges along the marginal (tangent) direction, and therefore its inverse converges to zero, asymptotically.

As shown in Figure 2c,d, the exact solution (Equation 44) of the Fokker–Planck equation is well reproduced by piecing together Gaussian tubes of covariance $Q^{-1}\left(t\right)$, each computed around a definite point of the noiseless limit cycle (the spurious lines orthogonal to the circle in Figure 2d are due to the finite sampling of the Gaussian tubes, that should ideally be a continuum).

As we may mostly be interested in the solution of the Lyapunov equation near unstable periodic orbits (like in a chaotic system), we then need to solve the same problem backwards in time, otherwise said by studying adjoint evolution, or the adjoint Lyapunov equation.

### 4.3. Adjoint Lyapunov Equation

The backward evolution is described by the adjoint Fokker–Planck equation
(46)$$\partial_{t}\rho \left(x,t\right)=v\left(x\right)\partial_{x}\rho \left(x,t\right)+\Delta \partial_{xx}\rho \left(x,t\right)\;.$$
Following the line of thought of Section 4.1, we can write the path-integral evolution of a Gaussian density in the neighborhood of an orbit
(47)$$\begin{array}{ccc} \rho_{a}\left(z_{a}\right) & = & \frac{1}{C_{a+1}}\int \left[dz_{a+1}\right]\;e^{-\frac{1}{2}\left(z_{a+1}-\left(1+A_{a}\delta t\right)z_{a}\right){}^{⊤}\frac{1}{\Delta \delta t}\;\left(z_{a+1}-\left(1+A_{a}\delta t\right)z_{a}\right)-\frac{1}{2}z{}_{a+1}^{⊤}\frac{1}{Q_{a+1}}\;z_{a+1}}\\ & = & \frac{1}{C_{a}}\;e^{-\frac{1}{2}z{}_{a}^{⊤}\frac{1}{Q_{a}}z_{a}}\;, \end{array}$$
where
(48)$$Q_{a}={\left(1+A_{a}\delta t\right)}^{-1}\left(Q_{a+1}+\Delta \delta t\right){\left[\left(1+A_{a}\delta t\right){}^{⊤}\right]}^{-1}\;.$$
Analogously to the forward evolution, we can take the limit of infinitesimal time intervals and get the differential equation
(49)$$\dot{Q}=\Delta -A\left(t\right)Q-QA{}^{⊤}\left(t\right)\;,$$
the adjoint (or backward-) Lyapunov equation. Compared to the forward Lyapunov Equation (Equation 34), the adjoint evolution (Equation 49) features the ‘time reversal’ operation A(t)→−A(t), and therefore we can still use Equation (Equation 37) as a solution, as long as the Jacobian along the orbit is computed as
(50)$$\frac{d}{dt}J\left(t,t_{0}\right)=-A\left(x\right)J\left(t,t_{0}\right),\;J\left(t_{0},t_{0}\right)=1\;,$$
and its computation follows the time reversed flow, that is the solution to the dynamical system $\dot{x}=-v\left(x\right)$.

## 5. Non-Circular Limit Cycles: Classical Noise vs. Quantum Dissipation

We now turn our attention to non-circular limit cycles with background noise, and determine the steady-state density distribution yielded by the Fokker–Planck equation. We do so by integrating the Lyapunov equation in the neighborhood of a trajectory that eventually converges to the limit cycle.

The paradigmatic models of our choice both come from the nonlinear oscillator
(51)$$\ddot{x}+\omega_{0}^{2}x+\mu \;g\left(x,\dot{x}\right)=0\;,$$
with *g*, a nonlinear function of position and velocity. Equation (Equation 51) may be reduced to a dynamical system as the Van der Pol oscillator, by taking $g=\left(x^{2}-b\right)\dot{x}$, or as the Rayleigh model, by taking $g=\frac{{\dot{x}}^{3}}{3}-b\dot{x}$ in Equation (Equation 51). In what follows, we shall set b=3, $\omega_{0}=1$, and we will tweak μ. The Van der Pol oscillator takes the form
(52)$$\begin{array}{ccc} \dot{x} & = & y\\ \dot{y} & = & -\mu \left(x^{2}-3\right)y-x\;, \end{array}$$
while the Rayleigh model reads (it is conventional to swap *x* and *y*)
(53)$$\begin{array}{ccc} \dot{x} & = & y-\mu \left(\frac{1}{3}x^{3}-3x\right)\\ \dot{y} & = & -x\;. \end{array}$$

In both classical systems, the dynamics converges to a limit-cycle of non-circular shape (Figure 3), which depends on the parameters, and it is characterized by fast and slow motions. One, therefore, expects the interplay of noise with the inherent nonlinear contraction to be non-uniform, unlike in the circular limit cycle examined in the previous section, and to give rise to a stationary density distribution of varying covariance along the cycle. We consider both models (Equation 52) and (Equation 53) for different values of the parameter μ, so as to gradually increase the eccentricity of the limit cycle, from a deformed circle (Figure 4a) to a nearly rectangular orbit (Figure 5b), where the deterministic stretching/contraction are most inhomogeneous along the cycle. The most notable feature is the oscillation of the covariance of the Gaussian solution of the linearized Fokker–Planck equation along the direction orthogonal to that of noiseless motion, denoted by σ in Figure 4c,d and Figure 5c,d: the more eccentric the limit cycle, the more widely and rapidly σ oscillates. That translates to a Gaussian steady-state density featuring a width that increasingly depends on the position along the orbit with the parameter μ, as we can see in the three-dimensional/density plots of Figure 4e,f and Figure 5e,f. As anticipated, these monodromic Gaussian distributions computed by means of the Lyapunov equation and portrayed in the figures would be, in a chaotic setting, the building blocks of a partition of the noisy phase space, whose non-uniform resolution is determined by their overlaps.

The Gaussian solutions of the Lyapunov equation computed and illustrated here share common traits with the steady-state Wigner function of the two oscillators (Equation 52) and (Equation 53), as obtained from a fully quantum mechanical computation that has recently appeared in the literature [25]. In that work, the quantization is performed by means of creation/annihilation operators
(54)$$\hat{a}=\frac{1}{2}\left(\hat{x}+i\hat{y}\right)\;,$$
and its adjoint ${\hat{a}}^{†}$, while dissipative terms are added to the Liouville–von Neumann Equation (Equation 4), in the spirit of Lindblad’s formalism (in units of *ℏ*): (55)$$\rho_{t}=-i\left[\hat{H},\rho \right]+\sum_{j}\alpha_{j}D\left[f_{j}\left({\hat{a}}^{†},\hat{a}\right)\right]\rho \;,$$
where
(56)$$D\left[\hat{c}\right]\rho =\hat{c}\rho {\hat{c}}^{†}-\frac{1}{2}{\hat{c}}^{†}\hat{c}\rho -\frac{1}{2}\rho {\hat{c}}^{†}\hat{c}\;,$$
while the coefficients $\alpha_{j}$ are functions of the parameters $\omega_{0}$ and μ. The Hermitian Hamiltonian $\hat{H}$ has two distinct expressions for the Rayleigh and the Van der Pol oscillators, and, like $f_{j}$, it is a function of linear (e.g., ${\hat{a}}^{†}$), bilinear (${\hat{a}}^{†}\hat{a}$), and nonlinear (${\hat{a}}^{2}$) terms involving the creation/annihilation operators (see ref. [25] for details). The above Equation (Equation 55) was numerically integrated, and the Wigner function was then found to eventually concentrate around the classical limit cycles, that feature similar eccentricities to the ones considered in the present work and plotted in Figure 4 and Figure 5. In particular (Figure 6), the steady-state Wigner distribution is enhanced along ‘tubes’ of varying width, as it can be noticed in the more eccentric density plots of the Rayleigh model (Figure 6c,d). This feature is especially apparent in Figure 6d, where the high-density region is narrower along the vertical segments of the limit cycle (faster classical motion), and wider along its horizontal segments (slower classical motion). It compares directly with the Gaussian solution of the Lyapunov equation portrayed in Figure 5f.

It is noted that in the cited work, the authors did not integrate the Wigner equation, but the Lindblad equation with a full-fledged quantum-mechanical algorithm. In particular, the localization of the Wigner function around the classical limit cycles is not to be taken for granted, and it legitimates the parallel between their steady-state solutions and the local Gaussian tubes obtained in the present work from the noisy classical system.

On the other hand, the numerical steady-state solutions to Equation (Equation 56) obtained by the authors of [25] are clearly not Gaussians centered at the limit cycles (except in Figure 6a, the case of least eccentricity), as demonstrated by their varying intensities along the periodic orbits. In that sense, the Gaussian Ansatz that turns the Fokker–Planck into the Lyapunov equation carries limited information on the phase-space density distributions at equilibrium. Therefore, the analogy proposed here should be taken with a grain of salt, and only considered as a hint for the noisy-classical to quantum-dissipative correspondence in a particular system with nontrivial interplay of contraction and diffusion.

Finally, we would like to briefly comment on the difference between classical and quantum dissipation in the present models featuring limit cycles. In the classical system the dissipation produces damping, which is then balanced by the noise-induced diffusion. Instead, the quantum dissipation, generated by the characteristic Lindblad terms in Equation (Equation 55), is responsible for both the ‘friction’ that drives densities to localize along the classical limit cycles, and the diffusion that spreads out the steady-state Wigner density distribution in the same region of the attracting orbits. This is consistent with the more general picture of Section 2.2, where the quantum dissipation brings about both a damping and a diffusive term in the Wigner equation.

## 6. Summary and Discussion

Having reviewed the parallels between the problem of dynamical evolution of a quantum system subject to dissipation and that of a stochastic process ruled by the Fokker–Planck equation, we have narrowed our attention down to chaos, and, in particular, to the problem of an inherent scale resolution of the phase space. The issue is measuring the conjugated variables down to a certain precision, which may be set by the balance of the contraction rate of the classical chaos with the coupling to the environment, the source of dissipation.

Using the analogy with the problem of classical chaotic dynamics with background noise, we consider the Fokker–Planck equation, and study its local solutions in the neighborhood of a periodic orbit, that effectively give the latter a finite width, in the phase space. Solving the problem for two-dimensional limit cycles, as done here, is the starting point: in a chaotic setting, a number of periodic tubes of finite width that proliferate exponentially with their length must end up overlapping, and thus determine the finest resolution for the noisy/quantized state space, that is expected to be non-uniform, as chaos interacts differently with diffusion/dissipation everywhere, in general.

The analysis performed in this venue shows that the problem is tractable, and it provides the basic technology to attack it. Complications and obstacles are ahead for higher-dimensional systems, where stable, unstable, and marginal directions coexist along the same orbit, and where the solution to the adjoint Fokker–Planck operator introduced here will almost certainly be instrumental to the method. Still, the progress already achieved by periodic orbit theory in such complex models as the Kuramoto–Sivashinsky or the Navier–Stokes equation gives us confidence in the feasibility of the optimal partition hypothesis in higher-dimensional chaos.

## Figures and Tables

**Figure 1 entropy-25-00411-f001:**
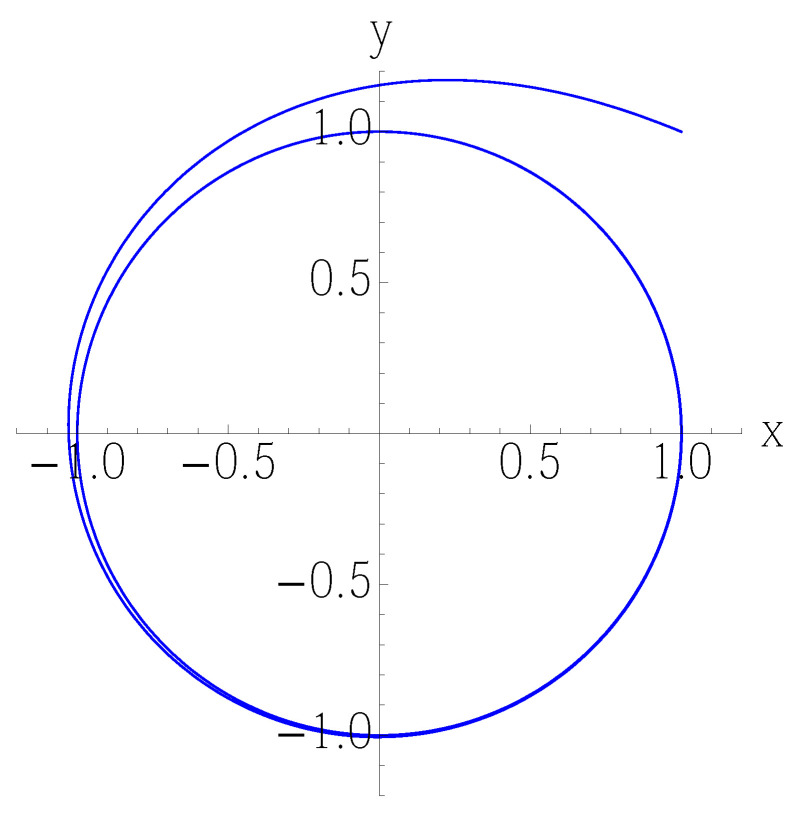
Solution of the numerically integrated Equation (Equation 38) without noise. Any initial condition converges to the circular limit cycle.

**Figure 2 entropy-25-00411-f002:**
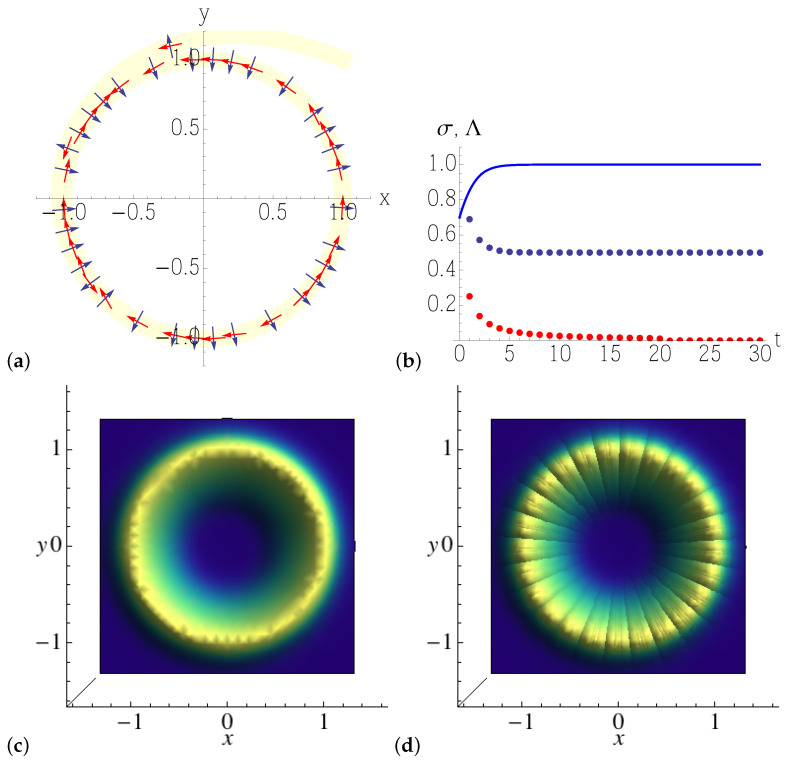
(**a**) Solution of the numerically integrated Equation (Equation 38), together with the eigenvectors (arrows) of the covariant matrix $Q^{-1}$, as given by the solution (Equation 37) of the forward Lyapunov equation, for noise amplitude 2D=0.1. The light yellow stripe is a pictorial representation of the Gaussian tube along the limit cycle (plan view). (**b**) The eigenvalues $\Lambda_{1}$ (blue dots), $\Lambda_{2}$ (red dots) of $Q^{-1}$, and the width σ (solid line) of the evolved density versus time *t*. (**c**) The exact steady-state solution (Equation 44) to the Fokker–Planck equation for a noisy circular limit cycle. (**d**) The approximation to the same steady state, obtained by piecing together solutions (Equation 37) to the Lyapunov equation around the limit cycle.

**Figure 3 entropy-25-00411-f003:**
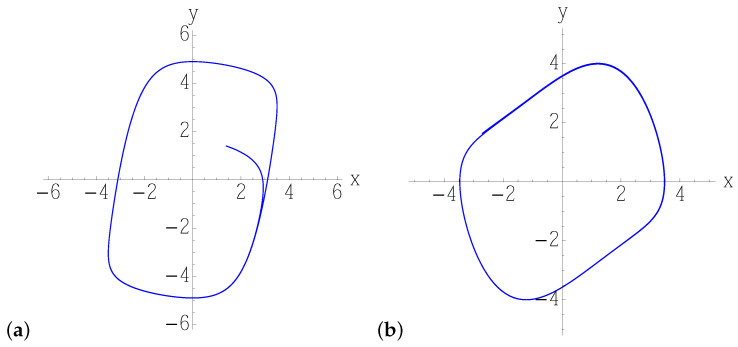
Solution of the numerically integrated (**a**) Equation (Equation 53), and (**b**) Equation (Equation 52), without noise. Any initial condition converges to a limit cycle.

**Figure 4 entropy-25-00411-f004:**
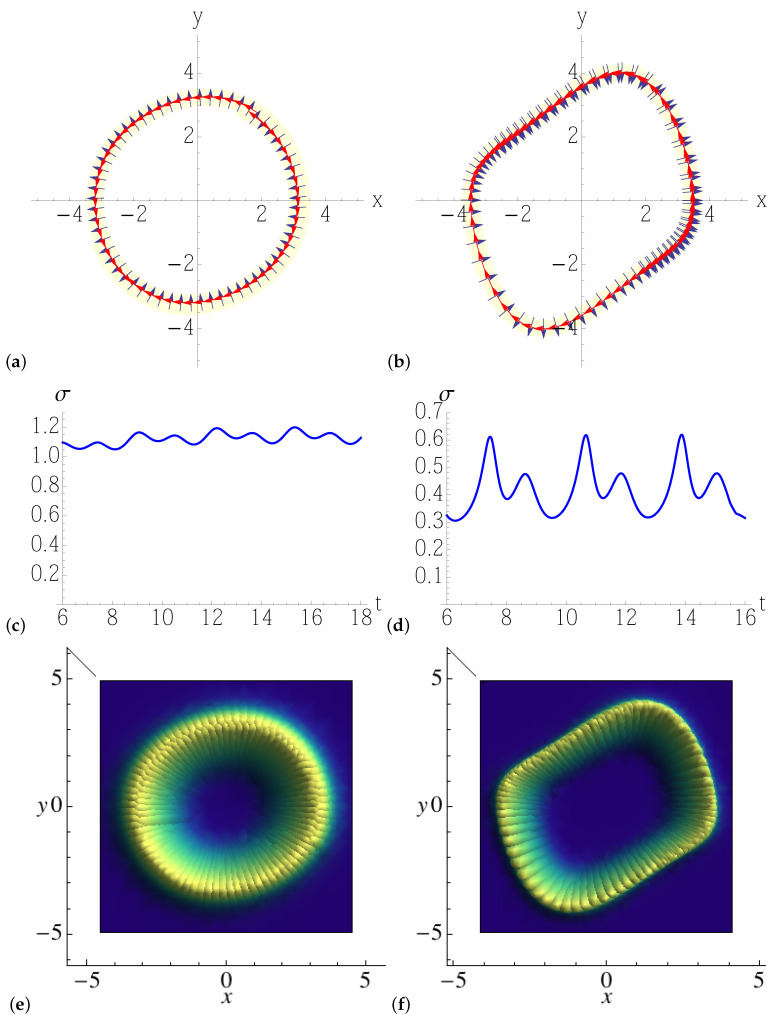
Top: solution of the numerically integrated Equation (Equation 52), together with the eigenvectors (arrows) of the covariant matrix $Q^{-1}$, as given by the solution (Equation 37) of the forward Lyapunov equation. In Equation (Equation 52), we take (**a**,**c**,**e**) μ=0.03, and (**b**,**d**,**f**) μ=0.2, while the amplitude of the noise is set to 2D=0.1. The light yellow stripe represents the width σ of the Gaussian density around the limit cycle. Middle: width σ of the Lyapunov tube, determined by the non-zero eigenvalue of $Q^{-1}\left(t\right)$ vs. time *t*. The cycle period is $t_{p}\approx 7$ time units. Bottom: Gaussian solutions of the linearized Fokker–Planck equation along the limit cycle.

**Figure 5 entropy-25-00411-f005:**
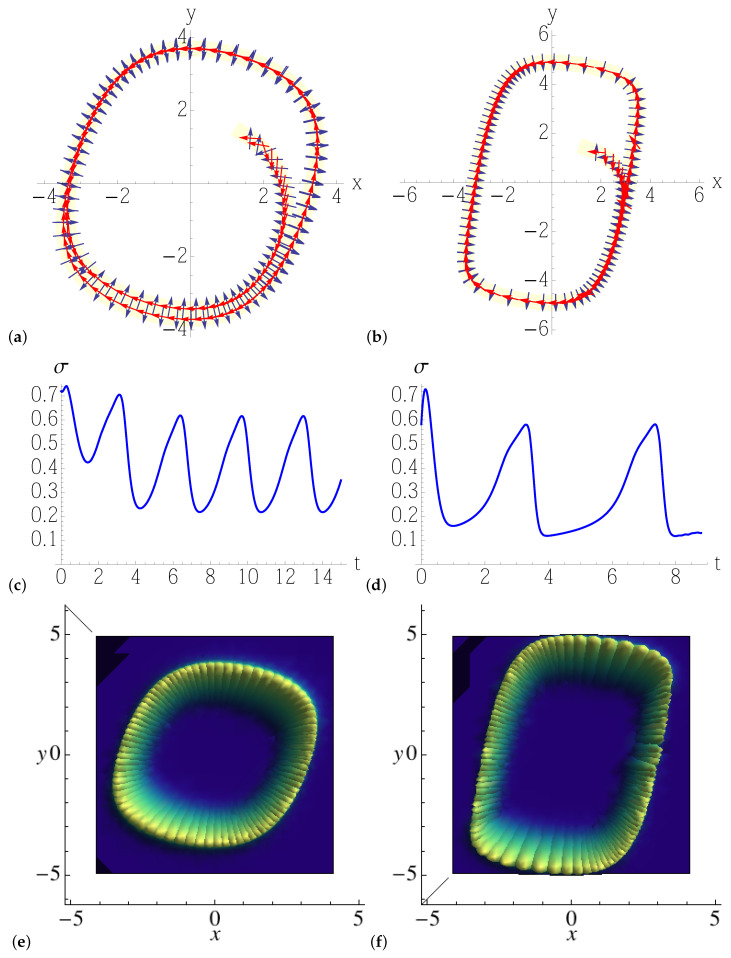
Top: solution of the numerically integrated Equation (Equation 53), together with the eigenvectors (arrows) of the covariant matrix $Q^{-1}$, as given by the solution (Equation 37) of the forward Lyapunov equation. In Equation (Equation 53), we take (**a**,**c**,**e**) μ=0.3, and (**b**,**d**,**f**) μ=0.8, while the amplitude of the noise is set to 2D=0.1. The light yellow stripe represents the width σ of the Gaussian density around the limit cycle. Middle: width σ of the Lyapunov tube, determined by the non-zero eigenvalue of $Q^{-1}\left(t\right)$ vs. time *t*. The cycle period is $t_{p}\approx 7$ time units. Bottom: Gaussian solutions of the linearized Fokker–Planck equation along the limit cycle.

**Figure 6 entropy-25-00411-f006:**
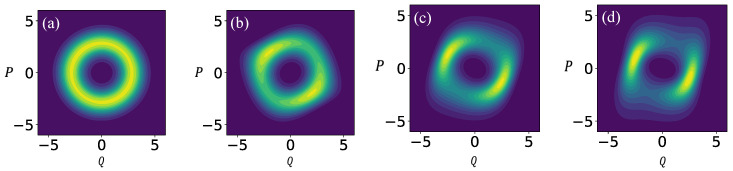
Quantum (**a**,**b**) Van der Pol and (**c**,**d**) Rayleigh oscillators from Reference [25]. The steady-state Wigner function (density plot) localizes around limit cycles of increasing eccentricity (controlled by the parameter μ in the classical models (Equation 52) and (Equation 53)). Courtesy of A. Chia.

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
