# Peer review of "Towards the Resolution of a Quantized Chaotic Phase-Space: The Interplay of Dynamics with Noise"

_entropy, 2023, doi:10.3390/e25030411_

Round 1
Reviewer 1 Report
The paper addresses the relation between a quantal and a classical description of equilibration in physical systems. It comes essentially in two parts. In Sections 1 to 3, an overview is given of existing or planned approaches to the problem. That part does not contain any novel results, in its reference to future work planned by the authors is highly speculative, and is partly incorrect. Eqs. (19) to (21) incorrectly contain the factor $W^{2s)$ which should be replaced by unity. In the absence of that factor, Eq.~(21) constrains $t^*$ in terms of $s$-fold derivatives of the potential $V$. It is not clear for which value of $s$ Eq. (21) is supposed to hold, and why. All that material has very little connection with or bearing on the developments in Sections 4 and 5. It overloads the paper and exhausts the reader before he/she arrives at that part of the paper (Sections 4 and 5) that contains new material. These Sections deal with few-degrees-of-freedom systems and are, thus, far removed from the ultimate goal of the authors (equilibration in quantum many-body systems). Nevertheless these results are a first step in the desired direction and deserve publication.
I suggest that the paper be returned to the authors for a thorough revision. Sections 1 to 3 should be shortened to maximally 2 pages, focusing on what is to follow in what are now Sections 4 and 5. Section 6 should be reworded accordingly. Some of the references can be dropped.
Author Response
Response to Reviewer 1
We thank Reviewer 1 for their thorough and careful reading, and for pointing out errors, which we have corrected in the present revision. In particular, the spurious extra factor $W^{2s}$ appearing in equations (19)-(21) has now been removed, and we have specified below Eq. (21) that the negligible higher-order terms of the Wigner equation are those for $s$ greater or equal to unity, within the discussed timescale.
Concerning sections 1-3 of the manuscript, we understand that they may not resonate with an expert in the field of quantum-classical correspondence. Yet, Reviewer 2 maintains that the manuscript is “of considerable interest to newcomers” to the field, largely because of the informative background provided in the introductory sections. We too believe that providing enough background on the Wigner equation in chaos and dissipation does help put our original work into context, especially for the non-expert readers. It is also noted that sections 2-3 together take four pages (out of the 16 of the whole manuscript), not an excessive length for reviewing decades of progress, in our opinion. By virtue of the above considerations, as well as of the guidelines of the Special Issue “Quantum Chaos” we are submitting our contribution to, we wish to leave the exposition unchanged in the present revision.
Reviewer 2 Report
The work by Lippolis an Shudo contributes to the fundamental problem of quantum-classical correspondence. The paper consists of two parts where the first part can be viewed as an extensive introduction to the problem. In particular, the authors explains in details how the effect of environment can reduce the quantum equation for the Wigner function to the classical Fokker-Planck equation.
In the second part of the work the authors discuss an approximate method of solving the Fokker-Planck equation in a particular case where the dynamics of the classical system is a limit cycle. Finally, they compare the obtained solutions with the stationary solution of the master equation borrowed from Ref.[25]. A qualitative agreement is noticed.
The paper is of some interest to experts and of considerable interest to newcomers to the field. However, before publishing the manuscript the authors should correct a few typos and clarify some points:
1. Eqs. (8) and (9) are not consistent because $\psi(…)\psi^*(…)$ is a density matrix.
2. $c_s(-\hbar^2)$ in Eq.(13) is not specified in the text.
3. Eq.(31) gives the impression that one is dealing with 1D case, which contradicts with Eq.(38) where one finds $x$ and $y$.
4. There is something wrong with Eq.(51) because Eqs.(52),(53) do not follow from this equation.
5. Line 315: "blocs" or "blocks"?
Author Response
Response to Reviewer 2
We greatly appreciate the Referee's feedback on our manuscript. We have made the suggested corrections in the present revision as follows:
-
Eqs. (8) and (9) are now consistent with the definition of Weyl transform
-
We have specified the value of $c_s$ in Eq. (13)
-
We have modified the notation in section 4, so that x vector is now in bold character.
-
Indeed the definitions of the oscillators given before were incorrect, now properly fixed in Eqs. (51)-(53)
-
“blocks” is the standard form, now used throughout the manuscript
Round 2
Reviewer 1 Report
The authors have followed my recommendations. I recommend publication.